# Knowledge and practices of birth preparedness and complication readiness among pregnant women in Eastern Province, Zambia: A qualitative study

Muyereka Nyirenda[1]*, Choolwe Jacobs[1], Mpundu Makasa[2], Alice Ngoma Hazemba[2]

**1** Department of Epidemiology and Biostatistics, School of Public Health, University of Zambia, Lusaka, Zambia, **2** Department of Community and Family Medicine, School of Public Health, University of Zambia, Lusaka, Zambia

* muyereka@gmail.com

## Abstract

Birth preparedness and complication readiness are key strategies for reducing maternal and neonatal mortality. This study aimed to explore the knowledge and practices of birth preparedness and complication readiness among pregnant women attending antenatal care in selected health facilities, using qualitative insights to identify barriers, facilitators and cultural factors. A phenomenological qualitative approach was used to explore pregnant women's knowledge and experiences of birth preparedness and complication readiness. Participants were recruited through convenience sampling, and a total of seven focus group discussions (FGDs) were conducted across seven health facilities, involving 53 participants. Data collection took place in July 2023, and the transcripts were systematically analyzed using NVivo software to identify key themes and patterns emerging from participants' narratives. Four identified themes were knowledge of birth preparedness and complication readiness; knowledge of danger signs of pregnancy; practices of birth preparedness and major delays to seek care. Participants listed common labour and delivery requirements. However, for complication readiness, most of them knew the types of complications but had little knowledge about preparation for such complications. Challenges such as lack of money to buy birth requirements and inadequate partner support led to poor preparations for pregnancy and childbirth. The use of traditional medication to hasten labour negatively influenced early care seeking. Distance to health facilities and lack of transport delayed access to healthcare. We found that pregnant women understood labour requirements but lacked knowledge on complication readiness. Financial constraints, insufficient partner support, reliance on traditional medicine, and long distances to health facilities hindered preparedness. This highlights the need for education, partner support and accessible healthcare.

**Data availability statement:** All data underlying the findings in the manuscript are provided within the manuscript and uploaded as supplementary information.

**Funding:** This study was funded by NORHED-PRICE, Grant Number 70324 only for data collection and analysis. NORHED PRICE stands for NORwegian programme for capacity development in Higher Education and research Development (NORHED) PRimary health care Institutional capacity building Curriculum development and leadership training Education system strengthening (PRICE). However, the funder had no role in study design, manuscript preparation and publication.

**Competing interests:** The authors have declared that no competing interests exist.

## Introduction

Maternal mortality has remained a global public health challenge for many years [1]. In 2020, an estimated 287,000 women globally died from a maternal cause [2]. In the same year, the global maternal mortality ratio (MMR) was estimated at 223 per 100,000 live births while that of sub-Saharan Africa stood at 545 per 100,000 live births (WHO, 2023). Zambia has a high MMR of 278 per 100,000 live births and a high stillbirth rate (SBR) of 15.9 per 1000 births [3,4].

Birth Preparedness and Complication Readiness (BP/CR) is an approach that aims at raising awareness and creating demand for quality antenatal care (ANC) services [5]. Recognizing its crucial role in reducing maternal and neonatal deaths, the World Health Organization (WHO) recommends and promotes BP/CR intervention as a fundamental component of ANC and the integrated management of pregnancy and childbirth [6]. A birth preparedness plan includes identification of place of birth, birth attendant, funds for birth-related and emergency expenses, birth companion, support at home while away and identification of compatible blood donors [7].

Many deaths are preventable and result from complications such as hemorrhage, infection, abortion and eclampsia in mothers, and asphyxia, infection and prematurity or low birth weight among the newborns [8]. Complications of pregnancy and childbirth are the leading causes of death and disability among women of reproductive age in developing countries [9].

In many societies in the world, cultural beliefs and lack of knowledge inhibit advance preparations for delivery [7]. When complications occur, the unprepared family will waste time in recognizing the problem, getting money, finding transport and reaching the appropriate health facility [10]. Knowledge of pregnant women on BP/CR improves recognition of problems in pregnancy, reduces the delay in deciding to seek care and guides on appropriate sources of care, making the care seeking process more efficient [11].

A study done in Ethiopia showed that only a small proportion of women were prepared for birth and its complications [10]. This finding was not consistent with the results of the study done in Uganda [12]. Studies done in Uganda, Nigeria and Cameroon showed that knowledge of danger signs is associated with the practice of BP/CR [13,14]. Many of the studies that have been done have taken a quantitative approach with focus on determining the level of awareness and factors associated with BP/CR [6,10].

In spite of the benefits of BP/CR in the improvement of maternal and newborn health, there are no documented studies about BP/CR in Zambia. This study therefore aimed at exploring the knowledge and practices of BP/CR among pregnant women attending ANC in selected facilities of Chipata, Chadiza and Katete districts of Eastern Province. It was a pre-implementation study nested in the *Implementation Research on the WHO Antenatal Care Guidelines adapted to Country context* [15]. The results of this study will provide valuable information for design of programs and interventions to improve maternal and neonatal health.

Global Public Health
PLOS

## Methodology

### Ethics statement

Ethical clearance was obtained from Excellence in Research Ethics and Science Converge; (reference number **2023-Jan-018)** and the National Health Research Authority (reference number **NHRA 00002/12/04/2023)**. Permission to nest this study in the *Implementation Research on the WHO Antenatal Care Guidelines* was granted by the Principal Investigator from Population Council Zambia. Clearance to collect data was obtained from the Ministry of Health, Eastern Province Health Office. Participants were informed of the study objectives, and they consented to be part of the study. All the participants signed the informed consent forms. Participants were informed that they were free to stop or refuse to participate in the study at any time without any consequences. Confidentiality was strictly maintained. "Clinical trial number: not applicable."

### Study design, study setting and study duration

A qualitative phenomenological approach was employed using focus group discussions (FGDs) to explore and understand the knowledge and practices of BP/CR among pregnant women attending antenatal care (ANC). The study was conducted in selected health facilities across Chipata, Chadiza, and Katete Districts in Zambia's Eastern Province. It was nested within an *implementation research study on the adaptation of the WHO Antenatal Care Guidelines to the country context* [15], ensuring alignment with the real-world setting in which the new ANC package was being implemented. This study was conducted in three of the fifteen districts of Eastern Province, Zambia—namely Chadiza, Chipata, and Katete. Data were collected from seven selected health facilities across these districts. Chadiza has a total of 25 health facilities, Chipata has 47, and Katete has 40. According to 2019 data, antenatal care (ANC) coverage in these districts was reported as 15,175 in Chipata, 11,016 in Katete, and 4,294 in Chadiza [15]. This study was conducted from 3 to 28 July 2023.

### Study population

The target population was pregnant women attending ANC in the health facilities participating in the *Implementation research of the WHO Antenatal Care Guidelines adapted to country context*

### Criteria of inclusion

- Willing and able to give informed consent
- Pregnant women who had attended a minimum of two ANC visits

### Criteria of exclusion

- Women attending ANC booking visit
- Pregnant women with major complications of pregnancy

### Research participants

A total of fifty-three (53) participants took part in seven focus group discussions (FGDs) conducted at the selected study sites. The focus groups comprised pregnant women attending antenatal care (ANC) services during the study period. Participants were within the reproductive age range, with the majority being married, having attained secondary-level education, and being unemployed. Convenience sampling was used to recruit participants who were readily available and willing to participate, which was appropriate given the study's exploratory nature.

## Participant recruitment and selection

Women were recruited using convenience sampling with assistance of MCH staff. Seven FGDs comprising seven to eight participants were conducted to reach saturation. Data saturation was reached when additional data no longer brought new insights, themes, or perspectives to the research questions. Reaching saturation was a key criterion for determining when to stop the focus group discussions.

## Data collection tool

A semi-structured topic guide was used to conduct the discussions. The topic guide was translated into Chichewa and back-translated into English to maintain the meanings.

## Data collection

The FGDs were conducted in Chichewa, a local language commonly spoken in the study districts. They captured the knowledge and practices of pregnant women who were attending ANC with respect to BP/CR. The discussions were audio-recorded using digital recorders, and their duration was between 1 hour and 1 hour 30 minutes.

## Quality control

The team had a minimum of nurse midwifery training, and were proficient in both English and the local language. They also received extensive training on ethics in research, data integrity and confidentiality, and had experience in conducting FGDs. All authors are experts in qualitative research. The research team exercised due diligence to ensure that the study was conducted according to its design.

## Data analysis

The research assistants and the first author transcribed all FGD audio recordings verbatim into Chichewa before translating them into English. These transcripts were then imported into NVivo 12, where they were systematically coded and organized into thematic areas. A deductive approach was employed to analyze recurring patterns by coding key concepts and main ideas. This iterative coding process facilitated the identification, analysis, and reporting of emerging themes and patterns. Content analysis was conducted both electronically and manually to index themes, with direct quotes selected to illustrate the broader themes and sub-themes.

## Results

### Socio-demographic characteristics of the study participants

Fifty-three (53) pregnant women participated in seven (7) focus group discussions (FGDs) on birth preparedness and complications readiness. The health facilities and number of participants in the FGDs were seven from Chipata hospital affiliated health centre (HAHC), eight from Kapata health centre (HC), eight from Jerusalem rural health centre (RHC) and seven from Gondar HC from Chipata district. The participants from Chadiza district were eight from Taferansoni RHC and seven from Chadiza RHC, and eight from Vlamukoko RHC in Katete district. The majority of the participants (43.3%) were aged between 26–30 years. Education attainment was good, with most (49.1%) of the participants having attained secondary education level (**Table 1**).

### Themes and subthemes

The findings from the FGDs were summarized into four (4) major themes, namely: Knowledge of BP/CR; knowledge of danger signs; practices of birth and complication preparation and health seeking behaviour. **Table 2** summarizes the themes and sub-themes of the study.

**Table 1. Socio-demographic characteristics of the study participants in Eastern Province, Zambia, 2023.**

| Thematic Area | Category | Number of Participants | Percentage |
|---|---|---|---|
| **Age** | 18–20 | 5 | 9.4 |
| | 21–25 | 11 | 20.8 |
| | 26–30 | 23 | 43.4 |
| | 31–35 | 10 | 18.9 |
| | 36–37 | 4 | 7.5 |
| **Education** | Never Attended School | 7 | 13.2 |
| | Primary Education | 17 | 32.1 |
| | Secondary Education | 26 | 49.1 |
| | Higher (Tertiary) | 3 | 5.7 |
| **Marital Status** | Married | 44 | 83.0 |
| | Divorced | 7 | 13.2 |
| | Never Married | 2 | 3.8 |
| **Employment Status** | Employed | 12 | 22.6 |
| | Self-Employed | 15 | 28.3 |
| | Unemployed | 26 | 49.1 |

**Table 2. Themes and subthemes for knowledge and practices of birth preparation and complication readiness, Eastern Province, Zambia, 2023.**

| |
|---|
| **Theme: Knowledge of BP/CR** |
| Subthemes |
| • *Birth preparedness* |
| • *Complication readiness* |
| • *Challenges of BP/CR* |
| **Theme: Knowledge of danger signs of pregnancy and delivery** |
| Subthemes |
| • *Danger signs during pregnancy* |
| • *Danger signs during delivery* |
| **Theme: Practices of birth and complications preparation** |
| Subthemes |
| • *Preparation for emergency transport* |
| • *Identification of blood donor* |
| • *Acceptability of the practice of BP/CR by pregnant women* |
| **Theme: Major delays that pregnant women experience** |
| Subthemes |
| • *Delay to seek care* |
| • *Delay to reach care* |
| • *Delay to receive care* |

**1.0. Knowledge on BP/CR.** Within the continuum of reproductive health care, ANC provides a platform for important healthcare functions, including health promotion, screening, prevention, diagnosis, treatment and/or referral. The results describe the information provided by pregnant women regarding BP/CR as discussed in the following sub-themes and their supporting participant's quotes.

**1.1. Birth Preparedness (BP):** On birth preparedness, the women were able to itemize the requirements for labour and delivery. Among the items that they were able to list were baby blanket, wrappers, cord clamps, black plastic, bathing tub, stockens, head sock and under garments.

*"There is a plastic, JIK (disinfectant), baby blanket, wrappers, then a bucket, bath tub and some clothes …laughs, and a baby towel … and something for you to use after giving birth" (R5: FGD Chadiza RHC).*

*"The first thing you need to buy is a baby blanket, then a bath tub and its bucket, then six gloves, then 4 wrappers" (R2: FGD Vlamukoko RHC).*

**1.2. Complication readiness (CR):** With respect to complication readiness, most of the women knew the types of complications but had little knowledge about preparation for such complications. Some women said that they did not prepare for complications of pregnancy and childbirth because they did not know when these complications would happen.

*"Aah, we do not prepare for those because you cannot know what will happen in future" (R2: FGD Chadiza RHC).*

*"No…laughs…we do not prepare...because we do not know what is inside" (R7: FGD Vlamukoko RHC).*

Some women talked about preparing for transport in all the FGDs. They said that they would keep some money for transport in case of an emergency. Others said that they would borrow money from someone when an emergency occurred.

*"Yes, it's a requirement for a person to prepare some money. When you see that you are almost due, you should not be completely without money. You need to keep some money, that time when labour starts you simply get that money, look for transport, and go to the hospital. It's a requirement whether you like it or not you have to keep it" (R2: FGD Chadiza RHC).*

**1.3. Challenges faced in Birth Preparedness and Complication Readiness (BP/CR):** Pregnant women face many challenges when it comes to BP/CR. These challenges can significantly affect their ability to access timely and appropriate maternal healthcare services and may increase the risk of maternal and neonatal complications, and death.

**1.3.1. Lack of money:** The biggest challenge that women face in managing birth preparation was the lack of money to buy baby layette as well as to prepare for transport. The lack of money was attributed to poor farming season. Others attributed the lack of money to not working or doing anything that would bring money. Most of the women depended on their partners for support.

*"Sometimes what causes us to fail to buy some things is because here in the village we do not do any business. So, we wait for husband to look for money. If he is also a drunkard when he finds money he will go to drink beer, that way preparation becomes difficult to fulfil. So, when we come here and are told to go and buy baby things and you find a man dillydallying in the village since it's hard for us to find money. Just what has happened this year, farming did not go well. That is why some of us we will not manage to prepare, due to lack of money" (R6: FGD Vlamukoko RHC).*

**1.3.2. Lack of partner support:** Partner support was not guaranteed in some of the women because some male partners did not show responsibility towards birth preparedness. Some women complained that some men would run away after making a woman pregnant, leaving her to fend for herself through piecework.

*"Laughs…just the way we are here, just after our husbands make us pregnant and then go away maybe to Katete (another district). It will be us suffering since we are pregnant, begging and kneeling in people's homes looking for*

*piece work with struggles. When you find one and you get a K20 ($1), you tremble [with joy] to go and buy baby socks and others, things go on just like that" (R3: FGD Chadiza RHC).*

**1.3.3. Results of inadequate preparations for pregnancy and childbirth:** When asked about what happens at the health center to pregnant women who go with incomplete items, some women reported that women who may not have adequately prepared for delivery opted to deliver from home for fear of being shouted at by healthcare providers.

*"Women have difficulties with birth preparations. Therefore, they decide to give birth from home because the staff (healthcare providers) shout at them here" (R5: FGD Gondar HC).*

Some participants sympathized with the health facility staff and said that going to the health facility with inadequate items required for childbirth put the healthcare providers in an awkward position. They also reported that some health facilities assist women who go with fewer items.

*"Because they also face a challenge when you come with incomplete items. You find that sometimes those things are not available here [health facility] to use on you. So that is why they also get annoyed" (R7: FGD Chadiza RHC).*

Some women reported that healthcare providers reminded them that items required at the health facility did not have to be newly bought things, as long as they carried the essential items for delivery.

*"People from the clinic tell us that we are not forcing you to look for new things. As long as you can have essential items like gloves, pin and so on, you can still come to the clinic. They will not shout at you that why didn't you buy new things. So, you are supposed to come to the clinic as long you have important items" (R3: FGD Vlamukoko RHC).*

**2. Knowledge on danger signs.** The knowledge of pregnant women on danger signs during pregnancy and childbirth is crucial for their ability to recognize potential complications early and seek timely medical care

**2.1. Danger signs during pregnancy:** BP/CR largely depended on women's awareness and knowledge of danger signs. The common danger signs during pregnancy as mentioned by participants were bleeding in pregnancy and having low blood levels (anaemia). They also mentioned that a woman should not have severe headache, swelling of feet (oedema) or severe lower abdominal pains.

*"You find that you are bleeding when you are not in labour" (R4: FGD Vlamukoko RHC).*

*"She is not supposed to lack blood, I don't know, maybe you say less blood...A pregnant woman should not suffer from severe headache" (R2: FGD Tafelansoni RHC).*

Other symptoms mentioned that suggest danger in pregnancy were convulsing, water breaking before labour begins, baby not moving in the womb and excessive vomiting. They also mentioned that women should not be sick during pregnancy.

*"If the baby is not moving in the womb, you need to quickly rush to the hospital. If you have persistent abdominal pains before labour starts, you need to go to the hospital" (R2: FGD Chadiza RHC).*

*"Excessive vomiting, poor appetite, fitting and water breaking. When you are sickly or coughing during pregnancy, you need to rush to the hospital" (R5: FGD Vlamukoko RHC).*

**2.2. Danger signs during delivery:** Across the seven facilities, women were able to list a number of danger signs during delivery. Some of the danger signs mentioned included the following: labour that does not progress, excessive bleeding and umbilical cord around the neck of the baby.

*"An umbilical cord is wrapped around the neck may strangle the baby and it may fail to come out during delivery"* (R4: FGD Vlamukoko RHC).

*"Maybe when you bleed excessively before or after giving birth. For that, they usually send you to the big hospital for further checkups"* (R5:FGD Chadiza RHC).

**3. Practices of birth preparedness and complication readiness.** BP/CR involves a range of practices and actions that pregnant women, their families, communities and healthcare providers can undertake to ensure a safe pregnancy, childbirth, and postpartum period. Emergency transport, identification of blood donor and acceptability of BP/CR practices emerged as crucial aspects to consider.

**3.1. Emergency transport:** Preparation for transport was one of the main health topics that the pregnant women were able to remember about birth preparedness. A number of women said that they considered transport as an important part of birth preparation. Others said that they would either keep the money or pay a taxi driver in advance.

*"It's a requirement to prepare transport money. It's just that we fail or maybe we do not pay attention. Since pregnancy complications do not have a specific time, you may say that I will prepare transport money when I reach 8 months, but you could be surprised that you start bleeding before that time. That's why when you are pregnant and you have a bit of money you need to keep it. If you cannot keep it, it is better to give a Taxi driver in advance. That way, if an emergency occurs you will simply make a phone call to the taxi driver. If there were no prior arrangements for transport you may find that your husband or family members do not have money when you urgently need it"* (R2: FGD Chadiza RHC).

**3.2. Identification of blood donor:** A question on the identification of a blood donor in case of an emergency requiring blood transfusion came as a surprise to most of the participants. Most of the women did not prepare a blood donor because the HIV status and blood type of the donor was not known. Most of them were not keen to have their relatives donate blood to them for fear of contracting HIV.

*"What if my mother and I have the same blood type but I do not know her HIV status? In case I need blood, would they give me her blood when she is HIV positive while I am negative? That is dangerous... Laughs"* (R1: FGD Chipata HAHC).

*"No, that doesn't happen because we do not know what will happen in future. When pregnant, we eat food required to increase the level of blood so that the body can have enough blood at the time of delivery. I cannot prepare for blood transfusion, what if my relative is sick? Blood groups also differ, so at the hospital they will know my blood type and the type of blood to give me. You cannot depend on your relative, what if they are infected with HIV?"* (R7: FGD Chadiza RHC).

A few participants knew their blood groups and had someone in mind who could donate blood to them. They said that they were told to prepare a relative in case they needed to be transfused.

*"I can get it from my mother and father…my mother is blood group B while my father is blood group O …. I know this because they got my blood to give to my mum when she was sick some years ago before I became pregnant"* (R8: FGD Tafelansoni RHC).

**3.3. Acceptability of the practice of BP/CR by pregnant women:** Despite the challenges women face on BP/CR, they still felt that it was necessary for them to prepare. Some women said that they prepare with difficulties and that they could only prepare for what they could afford. However, the understanding among pregnant women was that every pregnant woman needed to start preparing the moment they knew that they were pregnant.

*"Every pregnant woman begins to prepare the moment she knows that she is pregnant. She should continue to prepare bit by bit as she waits to reach nine months. So, a pregnant woman should have transport money and everything required for the baby. She should also quickly go to the hospital the moment labour starts, that way, complications will not be there"* (R7: FGD Chadiza RHC).

**4. Health seeking behavior.** Regarding health seeking behaviour, the participants spoke highly of the confidence that they had in the health system. All of them had the desire to attend antenatal care and to deliver from health facilities.

**4.1. Major delays that pregnant women experience:** For this study, women reported delays in seeking care, in reaching care and in receiving care. The following sub-themes provide the context in which the participants described the delays:

**4.1.1. Delay in seeking care:** Some women felt that delay in seeking care was deliberate as some women would be feeling the labour pains but still decide not to go to the health facility. Some women wait until delivery is eminent before they decide to go to the health facility. Other women said that they did not want to wait for a long time at the health facility.

*"What becomes a problem is when a person knows that labour pains have started but she is still at home, not coming to the hospital, as such time is running out. By the time she thinks of coming here, time would have gone"* (R3: FGD Tafelansoni RHC).

*"Yes, some women do not recognize labour signs. They may have lower abdominal pains or back pains but they may not know that labour that has started. That way, some women fail to recognize the problem because they believe that they have other signs when labour starts. It is for this reason that we should be going to the hospital immediately when we start experiencing any pain"* (R7: FGD Chipata HAHC).

Some women said that they sometimes delay leaving home because they would be "cooking" the pregnancy. This term was used across the study districts and it implied waiting until the labour is established. They said this would reduce time spent at the health facility. In this case, pregnant women were advised by parents especially grandparents to wait after labour pains start. The use of herbal medication as encouraged by parents contributed to this delay.

*"It happens, for some labour will start maybe at night just as they have said already that they wait for the pregnancy to cook so that they give birth as soon as they arrive at the health facility. They end up giving birth on the way to the health facility"* (R6: FGD Vlamukoko RHC).

*"Sometimes what we do when we are bleeding or fluid is coming out is that we will be seated at home. We also call grandma who will dig some roots and make us to drink. The grandma will also be busy with her things, instead of taking you to the hospital. The biggest problem is from the home"* (R5: FGD Chadiza RHC).

**4.1.2. Delay in reaching care:** The common reason for the delay to reach care was transport challenges due to lack of money. The participants felt that finding transport was easier when money was available.

*"Some delays are due to lack of transport money. Someone may not be able to walk because the legs are swollen. It is therefore important to save transport money"* (R4: FGD Chipata HAHC).

**4.1.3. Delay in receiving care:** Sometimes delay happened at the health centers where the healthcare providers took long to attend to clients or to identify complications and refer them to the higher level of care.

*"It happens, you arrive at the hospital but the nurses are busy. What should happen is that when you arrive, the nurse should hurry to attend to you. So that he or she sees if the baby is ok or not. But sometimes the one dillydallying is the nurse" (R2: FGD Vlamukoko RHC).*

*"That happens [number 6] … In my last pregnancy, I was told that she will come, she said I will give birth at 14:00 hours, and it was 08:00 hours. The moment the nurse left, the baby was born. When she came back, she said you forced the baby to be born early. It's should not be like that" (R6: FGD Vlamukoko RHC).*

Some women said that they sometimes delay going to higher level of care while at the lower level health facility because of the challenges in the referral system.

*"Like for the facilities here in the villages, delay is there because an ambulance will not be there to wait for a pregnant woman. So when they see the complication that is when they call the ambulance. For the ambulance to start off and reach here, it would take time. That is really the reason pregnant women die on the way to the hospital. The challenge is there in the villages in terms of transport" (R7: FGD Chadiza RHC).*

## Discussion

In this study, we aimed to explore and understand the knowledge and practices of BP/CR among pregnant women who were attending ANC in selected facilities of Chipata, Chadiza and Katete districts of the Eastern Province. The findings provide baseline information for the *Implementation Research on the WHO Antenatal Care Guidelines adapted to Country context* with respect to BP/CR [15]. WHO emphasizes the need for BP/CR as both a preparation strategy and an important element for the safety and health of mothers and newborns during pregnancy, childbirth and in the postpartum period [16]. One of the important functions of ANC is to educate pregnant women on BP/CR [17]. This study therefore provides evidence regarding the need to strengthen complication readiness in health education during ANC contacts. BP/CR is crucial in ensuring safe pregnancies, deliveries, and postnatal care by facilitating timely access to skilled care, reducing delays, empowering women, preventing and managing complications [18].

This study found that the participants had knowledge of birth preparedness and knew the things that they were supposed to prepare for early in pregnancy. Knowledge is supposed to translate into empowered pregnant women who can make prompt decisions regarding their care [19]. The study found that birth preparedness was one of the key topics that were emphasized during ANC in the study facilities. This is different from a study done in Uganda, which found that pregnant women did not have adequate knowledge in birth preparedness [20]. Knowledge in birth preparedness is important for pregnant women make informed decisions, plan for a safe delivery, and recognize potential complications early on.

The knowledge of complication readiness among the ANC attendees was low. Pregnant women with limited knowledge of complication readiness may feel less confident in advocating for their health needs, expressing concerns or seeking additional support from healthcare providers [17,21]. The lack of adequate preparations for complications of pregnancy makes women vulnerable to adverse maternal and fetal outcomes. Studies conducted in Ghana and Nigeria highlighted the importance of emphasizing complication readiness during health education [22]. Empowering pregnant women with information about potential complications, and how to recognize and address them is essential for promoting maternal and neonatal health.

Among the challenges that pregnant women faced regarding BP/CR was lack of money to buy baby layettes and other requirements. This is similar to the results of a systematic review on ANC attendance in sub-Saharan Africa, which found

that financial barriers, including lack of money and transportation, were major challenges for women in achieving BP/CR [23]. Pregnancy can be a challenging time, especially for those facing financial hardships. There is need to connect pregnant women with local community resources such as social cash transfer and nonprofit organizations that provide support to low-income families.

This study found that some pregnant women did not have enough partner support towards birth preparedness. The lack of partner support can have significant implications for BP/CR. It can potentially lead to increased stress, limited decision-making ability, and a higher risk of adverse pregnancy outcomes [22]. A study done by Bitew et al. found that pregnant women who received support from their partners during ANC were more likely to engage in birth preparedness activities, such as saving money for childbirth expenses, identifying transportation options, and planning for a skilled birth attendant [24]. It has been demonstrated that women who received support from their partners were more likely to recognize danger signs and seek medical care promptly in case of complications. This is similar to the findings of the study done in Ethiopia and in Zambia on male involvement in BP/CR [25]. There is need to strengthen partner support to pregnant women, including addressing social and emotional needs, to ensure optimal maternal and neonatal health.

Participants were able to articulate danger signs of pregnancy, labour and childbirth. Pregnant women who are knowledgeable about danger signs during pregnancy, labor, and childbirth tend to have better outcomes compared to those who are not aware. These findings are different from those of a study in the sub-region, which found that pregnant women were not knowledgeable about danger signs [26]. BP/CR encourages pregnant women to be informed of the danger signs during labour and childbirth, so that they can choose the preferred place to deliver, the birth attendant and arrange for transport [10]. Such preparations are dependent on the awareness and knowledge of danger signs. Pregnant women who are aware of danger signs are better prepared to protect themselves and their unborn babies [21].

The women placed a lot of importance on preparation for transport to take them to the health facility in case they experienced some complications during pregnancy. Studies indicate that when complications occur, the unprepared family would waste time in looking for money, finding transport and reaching the health facility [10]. BP/CR is not easy to achieve especially in developing countries, where few women identify transportation ahead of childbirth and few women put aside funds for transport in case of emergency [6,13]. Reserving money during pregnancy is essential for ensuring easy access to the health facility.

One of the components of BP/CR is the identification of a blood donor early in pregnancy in case of an emergency requiring blood transfusion [7,14,26]. Identification of a blood donor is one of the key components of the Zambia ANC Guidelines [27]. In this study, most of the women were not aware that they needed to identify a blood donor in advance. This is one of the topics that should be emphasized in health education during ANC. Identifying a blood donor during pregnancy plays a critical role in emergency preparedness, ensuring timely access to compatible blood in case of severe bleeding [7].

Further, knowledge of birth preparedness and complication readiness is essential in reducing delays to access care especially in low and middle income countries (LMIC) where services are not close to communities [28,29]. Our findings indicate that women reported a cultural practice of 'cooking' the pregnancy as one of the reasons for the delay in seeking care. 'Cooking' of pregnancy is the ingestion of herbal medication to expedite labour, so that they do not stay for a long time at the health facility. Studies have shown that such cultural practices contribute to late reporting to a health facility [7,10]. Consuming traditional medication to expedite labour and childbirth can pose several dangers and risks to both the mother and the unborn baby [30]. These dangers may culminate in maternal or neonatal death, or both maternal and neonatal death. There is need to focus on educating both men and women on the risks associated with delays in deciding to seek care during pregnancy and labour [31]. The main problem with the second delay was transport challenges due to lack of money. This finding is similar to that of the study done in Ethiopia which highlighted the negative impact of the lack of ambulances to pick patients in the community and inadequate utilization of public transport due to lack of resources [6]. Averting the second delay may involve improving road networks, providing transportation subsidies or vouchers, or

establishing ambulance services to pick women in the community [17]. Some women reported a delay in moving from the lower level to the higher-level facilities. This was attributed to the referral system where there is only one ambulance servicing more than twenty health facilities. This is similar to the findings of a study done in Tanzania relating to delays in the referral system [32]. There is need to strengthen the referral system between primary and secondary healthcare facilities to ensure timely access to appropriate levels of care.

Most of the women mentioned that they preferred to deliver at the health facility because they could receive help in case of an emergency. To mitigate transport challenges in accessing care, they opted to move to the mothers' waiting shelters that were built within the health facility grounds. This finding is similar to the study conducted in Ethiopia on how women choose their place of delivery [33]. In Zambia, the number of health facility deliveries has steadily increased over the years to eighty-four (84) percent. However, the likelihood of a woman delivering at a health facility decreases with each subsequent birth. Ninety-two (92) percent of first-order births were delivered in a health facility, compared to seventy-five (75) percent of sixth-order higher births [3]. Delivering at a health facility is essential for ensuring safe and effective childbirth [34]. It provides access to skilled care, emergency interventions, and essential postnatal support, thereby helping to protect the health and well-being of mothers and babies.

## Limitations

Findings from a qualitative study may not be directly applicable to other settings or contexts, as BP/CR practices and healthcare systems vary across provinces. Secondly, convenience sampling may limit the generalizability and reproducibility of the findings, as the sample may not fully represent the broader population of pregnant women in the region. Participants may also have provided responses they perceive as socially desirable rather than expressing their true opinions. However, this qualitative study may have provided valuable insights into participants' experiences and perceptions at a specific point in time, but it may not capture trends in BP/CR practices over time.

## Conclusion

This study found that while pregnant women in Eastern Province of Zambia have good knowledge of birth preparedness particularly regarding transport, delivery planning, and awareness of danger signs. However, there are significant gaps in their knowledge and practices related to complication readiness, such as identifying a blood donor or preparing for emergencies. Financial constraints, limited partner support, and harmful cultural practices also pose barriers to timely care-seeking. Although ANC services emphasize birth preparedness, more targeted health education is needed to address complication readiness, enhance male involvement, and improve emergency readiness. Strengthening community support systems, referral system, and transportation infrastructure is critical to ensure timely access to skilled care and hence improve maternal and neonatal outcomes.

## Recommendations

Based on the study findings, it is recommended to strengthen health education during antenatal care by emphasizing all components of birth preparedness and complication readiness, particularly the identification of a blood donor and recognition of danger signs. Efforts should be made to increase male involvement in maternal health to enhance partner support, while addressing financial barriers through social protection programs and community-based support. There is also a need for targeted community sensitization to discourage harmful cultural practices that delay timely care-seeking. Improving emergency transportation and referral systems by increasing ambulance availability and enhancing road infrastructure is essential to reduce delays in accessing care. Additionally, promoting the use of mothers' waiting shelters and providing targeted support for multiparous women can further improve maternal and neonatal outcomes.

## Acknowledgments

We appreciate the pregnant women who shared their perceptions and practices of BP/CR in this study, and research assistants from study sites. We also appreciate the support from the project team of the *Implementation research on the WHO Antenatal Care Guidelines adapted to country context: Enhanced quality service delivery through a digitalized module*.

## Author contributions

**Conceptualization:** Muyereka Nyirenda.

**Data curation:** Muyereka Nyirenda.

**Formal analysis:** Muyereka Nyirenda.

**Funding acquisition:** Muyereka Nyirenda.

**Investigation:** Muyereka Nyirenda.

**Methodology:** Muyereka Nyirenda, Mpundu Makasa, Alice Ngoma Hazemba.

**Supervision:** Choolwe Jacobs, Mpundu Makasa, Alice Ngoma Hazemba.

**Validation:** Choolwe Jacobs, Mpundu Makasa, Alice Ngoma Hazemba.

**Writing – original draft:** Muyereka Nyirenda.

**Writing – review & editing:** Choolwe Jacobs, Mpundu Makasa, Alice Ngoma Hazemba.

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
