## [Decision Letter · Decision Letter 0]

PGPH-D-25-00507

Knowledge and Practices of Birth Preparedness and Complication Readiness among Pregnant Women in Eastern Province, Zambia: A Qualitative Study

Dear Dr. Nyirenda,

Thank you for submitting your manuscript to PLOS Global Public Health. After careful consideration, we feel that it has merit but does not fully meet PLOS Global Public Health’s publication criteria as it currently stands. Therefore, we invite you to submit a revised version of the manuscript that addresses the points raised during the review process.

We look forward to receiving your revised manuscript.

Kind regards,

Nicola Hawley

Academic Editor

Journal Requirements:

Additional Editor Comments (if provided):

Reviewers' comments:

Reviewer's Responses to Questions

**Comments to the Author**

1. Does this manuscript meet PLOS Global Public Health’s publication criteria?

Reviewer #1: Yes

Reviewer #2: Yes

Reviewer #3: Yes

2. Has the statistical analysis been performed appropriately and rigorously?

Reviewer #1: Yes

Reviewer #2: Yes

Reviewer #3: Yes

3. Have the authors made all data underlying the findings in their manuscript fully available (please refer to the Data Availability Statement at the start of the manuscript PDF file)?

Reviewer #1: Yes

Reviewer #2: Yes

Reviewer #3: Yes

4. Is the manuscript presented in an intelligible fashion and written in standard English?

Reviewer #1: Yes

Reviewer #2: Yes

Reviewer #3: Yes

Reviewer #1: This study is well-conducted, and the researchers are commended for their work thorough evaluating Knowledge and Practices of Birth Preparedness and Complication Readiness among

Pregnant Women in Eastern Province, Zambia: A Qualitative Study. The study is effectively structured, with a clear identification of the impact of Knowledge and Practices of Birth Preparedness and Complication Readiness among Pregnant Women in Eastern Province, Zambia for public health interventions. However, to enhance its credibility and ensure clear communication of the findings, the researchers should address the minor comments and suggestions provided above. Once these revisions are made, the study will be ready for publication and will provide valuable insights into decrease maternal complications and morbidity.

Reviewer #2: Strengths:

Topical Relevance:

The study addresses a critical public health issue aligned with global goals such as SDG 3 (Good Health and Well-being), particularly in maternal and neonatal health.

Methodological Appropriateness:

The use of a phenomenological qualitative approach is well-suited to explore personal knowledge, cultural norms, and experiences in-depth.

Data Collection and Analysis:

Conducting seven FGDs across seven facilities adds geographic and demographic diversity to the perspectives.

Use of NVivo software for thematic analysis suggests a structured and systematic approach to data analysis.

Identification of Clear Themes:

The paper identifies key themes that are highly relevant to BPCR and contextual barriers—knowledge gaps, cultural practices, financial constraints, and health system access.

Practical Implications:

The findings are not only informative but actionable—pointing towards the need for educational interventions, male involvement, and health system strengthening.

Areas for Improvement:

Clarify Sampling Rationale:

The use of convenience sampling should be acknowledged more transparently with a brief justification of how it may affect transferability of findings.

Participant Description:

Include demographic details (e.g., age range, parity, education) of the 53 participants to give readers a better sense of who was involved.

Language Use:

Some sentences are repetitive or wordy. For example:

“Pregnant women were able to list the requirements for labour and delivery...”

could be shortened to

“Participants listed common labour and delivery requirements...”

Depth of Insight into Themes:

While key themes are outlined, briefly highlighting quotes or subthemes would enrich the abstract. For example, include a notable insight on how traditional medicine delays care.

Cultural Context:

You mention “cultural factors” in the aim but don’t expand on these in the findings. It would strengthen the abstract to briefly note specific cultural beliefs or practices influencing BPCR.

Reviewer #3: Peer Review: Knowledge and Practices of Birth Preparedness and Complication Readiness among Pregnant Women in Eastern Province, Zambia: A Qualitative Study

Overall Assessment

This qualitative study explores birth preparedness and complication readiness (BP/CR) among pregnant women in Zambia’s Eastern Province. The research addresses an important gap in maternal health literature from the Zambian context and provides valuable insights into barriers and facilitators of BP/CR practices.

Introduction

• Well written with appropriate background and

• Clear study objectives.

Methodology

The subsection on study design, study setting, and study duration. The selection of the study site based on the nestedness within an implementation research study on the adaptation of the WHO Antenatal Care Guidelines seems adequate. However, the authors should elaborate on what these guidelines are and why it is important to select the study locations from the centers participating in this program. It is confusing how this relates to the justification for the number of stillbirths. This should be clarified.

The subsection on participant recruitment and selection should elaborate on the selection of women across different health facilities as described in the sociodemographic subsection of the results section.

Clarify the inclusion and exclusion criteria involving attending ANC visits, as it is confusing to readers. The authors should elaborate on these criteria.

Authors should elaborate on the justification for why saturation was reached at 7 FGDs

In the data analysis section, the transition from deductive to thematic analysis could be better explained

• Line 109: "Seven FGDs comprising seven to eight participants" - this suggests 49-56 participants, but 53 were reported. Clarify the exact composition.

Result

The authors provided no information on the link between the sociodemographic factors and the results/findings. Readers would be curious to know if the BP/CR is related to sociodemographic factors.

Discussion

• Line 434: The term “cooking pregnancy” needs more cultural contextualization.

• Comparison studies: Authors should focus on the most relevant studies, especially systematic reviews

• Limitations: The limitation section is brief and could address more methodological concerns.

Minor Technical issues

• Generally well-written, but some grammatical errors need correction

• Inconsistent abbreviation usage (BP/CR vs Birth Preparedness and Complication Readiness)

• Some sentences are overly long and could be simplified for easy readability.

References

• Generally appropriate and current

• Some formatting inconsistencies and corrections

**Do you want your identity to be public for this peer review?** For information about this choice, including consent withdrawal, please see our Privacy Policy

Reviewer #1: **Yes: ** Full name: Habtamu Molla Ayele

First name: Habtamu Molla

Last name: Ayele

Reviewer #2: No

Reviewer #3: **Yes: ** Queen Esther Adeyemo

---

## [Decision Letter · Decision Letter 1]

Knowledge and Practices of Birth Preparedness and Complication Readiness among Pregnant Women in Eastern Province, Zambia: A Qualitative Study

PGPH-D-25-00507R1

Dear Dr Nyirenda,

We are pleased to inform you that your manuscript 'Knowledge and Practices of Birth Preparedness and Complication Readiness among Pregnant Women in Eastern Province, Zambia: A Qualitative Study' has been provisionally accepted for publication in PLOS Global Public Health.

Best regards,

Nicola Hawley

Academic Editor

Reviewer Comments (if any, and for reference):

Reviewer's Responses to Questions

**Comments to the Author**

Reviewer #1: All comments have been addressed

Reviewer #3: All comments have been addressed

publication criteria?

Reviewer #1: Yes

Reviewer #3: Yes

3. Has the statistical analysis been performed appropriately and rigorously?

Reviewer #1: Yes

Reviewer #3: Yes

4. Have the authors made all data underlying the findings in their manuscript fully available (please refer to the Data Availability Statement at the start of the manuscript PDF file)?

Reviewer #1: Yes

Reviewer #3: Yes

5. Is the manuscript presented in an intelligible fashion and written in standard English?

Reviewer #1: Yes

Reviewer #3: Yes

Reviewer #1: The comments are purely addressed

Reviewer #3: This manuscript has shown considerable improvement from previous versions and addresses a critical gap in maternal health research in Zambia. The qualitative approach is appropriate and the findings provide valuable insights for policy and practice.

**Do you want your identity to be public for this peer review?** For information about this choice, including consent withdrawal, please see our Privacy Policy

Reviewer #1: **Yes: ** Full name: Habtamu Molla Ayele

First name: Habtamu Molla

Last name: Ayele

Affiliation:

Maternal and Child Health Directorate, Federal Ministry of Health, Addis Ababa, Ethiopia

Reviewer #3: **Yes: ** Queen Esther Adeyemo
